# Associations of Sleep and Health Functioning with Premature Exit from Work: A Cohort Study with a Methodological Emphasis

**DOI:** 10.3390/ijerph18041725

**Published:** 2021-02-10

**Authors:** Erkki Kronholm, Nathaniel S. Marshall, Minna Mänty, Jouni Lahti, Eero Lahelma, Olli Pietiläinen, Ossi Rahkonen, Tea Lallukka

**Affiliations:** 1Finnish Institute of Occupational Health, P.O. Box 40, 00032 Helsinki, Finland; erkki.kronholm@utu.fi; 2Sydney Nursing School, University of Sydney, Sydney, NSW 2006, Australia; nathaniel.marshall@sydney.edu.au; 3NHMRC Centre for Integrated Research and Understanding of Sleep (CIRUS), Woolcock Institute of Medical Research, 431 Glebe Pt Rd., Glebe, NSW 2036, Australia; 4Department of Public Health, University of Helsinki, Tukholmankatu 8B, P.O. Box 20, 00014 Helsinki, Finland; minna.manty@vantaa.fi (M.M.); jouni.mm.lahti@helsinki.fi (J.L.); eero.lahelma@helsinki.fi (E.L.); olli.k.pietilainen@helsinki.fi (O.P.); ossi.rahkonen@helsinki.fi (O.R.); 5City of Vantaa, Unit of Statistics and Research, Asematie 7, 01300 Vantaa, Finland

**Keywords:** premature retirement, sleep problems, health functioning, insomnia, epidemiology

## Abstract

Sleep and functioning are associated with a risk of early workforce exit. However, patterns of change in sleep and functioning through time have not been investigated using person-oriented approaches to show what features of sleep and functioning are associated with an early exit. We examined the pattern of interactions between sleep and health functioning characterizing homogenous subgroups of employees and their associations with premature work exit. An additional aim was to provide a tutorial providing detailed description on how to apply these models, compared to traditional variable based risk factors. We analyzed data from 5148 midlife employees of the City of Helsinki, Finland, surveyed over three phases (2000–02, 2007, and 2012). Using repeated measures latent class analyses (RMLCA) we classified people into groups based on their trajectories in sleep and functioning. We identified four longitudinal groups: (1) Stable good sleep and functioning (reference), (2) Persistent sleep problems and good or moderate functioning, (3) Poor functioning with good sleep, and (4) Problematic sleep and health functioning. Compared to group 1, elevated risk was found in all classes with group 4 being the worst. In conclusion, focusing on person-orientated patterns of interactions between sleep and functioning helped produce qualitatively different and quantitatively stronger predictions than using conventional risk factor methodology. Thus, longitudinal person-oriented approaches may be a more powerful method for quantifying the role of sleep and health functioning as risks for premature exit from work.

## 1. Introduction

Premature exit from work creates serious health and societal burdens in modern societies, including Finland [1]. Therefore, health and welfare social policies aim to extend work careers and raise the retirement age [2]. A number of factors, including health functioning as well as poor sleep, are associated with early exit from work and work disability. Several studies suggest that sleep problems have adverse impacts on work productivity and disability by increasing the risk of future sickness absence and permanent disability retirement due to declines in health and functioning [3,4,5,6,7]. Disturbed sleep due to thoughts of work and fatigue have also been found to be overlapping predictors of lack of return to work from long, and intermediate, term sickness absence [7]. In addition, sickness allowance due to mental disorders are strongly associated with disability retirement due to mental disorders [8,9]. Sickness allowance due to somatic conditions is a weaker risk factor [8]. In a Finnish study, all eight of the Short Form 36 Health Survey (SF-36) subscales and both summary scales [10] were associated with the occurrence of long sickness absence over the three-year follow-up period [11]. Consequently, reports of poor mental and physical health functioning provide early background risk markers for permanent impairment leading to an early exit from the workforce.

One issue with all of these studies is that they have employed variable oriented statistical approaches [12]. This creates difficulties in translating their results into single individuals [13] and may lead us to treating risk factors rather than identifying vulnerable individuals for public health interventions or clinical or public policy trials [13].

Sleep can be characterized in numerous ways and it’s not clear which sleep problems, and in what way, are associated with premature exit from work. Moreover, it is unclear whether the evolution of these problems through time predicts the development of work disability. Findings from studies into the role of sleep duration are somewhat equivocal [5,6]. The association with insomnia-related symptoms is more robust [5,14], but the studies have applied a variable orientated approach, without focusing on the patterns of change over time. An Australian study found that both clinical sleep disorders (obstructive sleep apnea, OSA, and insomnia) and sleep disturbance such as not getting adequate sleep, use of sleep aids, daytime symptoms, and excessive sleepiness were all independently associated with sickness absence [15]. The authors concluded that the relationship between sleep and sickness absenteeism encompasses more than the impact of clinical sleep disorders and that non-clinical sleep problems are causing the association. In addition, the relationship is found with brief, recurrent, and prolonged periods of sickness absence [15]. 

As such we decided to apply a person-oriented approach where individual development in sleep and functioning is seen as a process characterized by states that change through time [13] in a well characterized cohort with good repeated measurements of sleep, functioning, and exit from work. In this process different sleep problems and poor health functioning are seen as patterns of operating factors where each factor derives its meaning from its association to the others through time [13]. Such an approach thus extends from previous studies by focusing on patterns of change in sleep and functioning jointly, as well by applying person-oriented methods.

Consequently, this study had two aims, being partly an investigational study and partly a tutorial on the general approach. The first aim of this study was to examine the 10-year pattern of interactions between sleep and health functioning to characterize subgroups of employees and to determine whether the risk for premature exit from paid employment differs between these subgroups. Our second, additional aim was to provide more details of the method, using our study as an example of person-oriented approaches to RMLCA models to complement and to aid their understanding about when these might be employed in future studies. While the method has been used in other research fields RMLCA methods appear to have been very seldom used in sleep or insomnia research.

## 2. Materials and Methods

### 2.1. Participants

This study is part of the ongoing Helsinki Health Study (HHS) focusing on health and well-being, including health related functioning among employees of the City of Helsinki, Finland [16,17]. Inclusion criteria for this study were participation in survey questionnaires at all three consecutive phases of the study in 2000–02, 2007, and 2012, and providing information for all variables of interest in repeated surveys. There were 8960 people (7168 women and 1792 men) who responded at the first study phase when all of them were employed by the City of Helsinki. At the second phase, 7332 people responded to the survey (5980 women and 1352 men) when ~70% of them were still working. At the third phase, 6808 people responded (5558 women and 1250 men) when ~50% of them were still working. Those participants who provided full information on key independent variables (sleep-related symptoms and health functioning) from all three study phases were included in our analyses (*n* = 5148 people; 4204 women and 944 men). Except for missing data there were no exclusion criteria, as all participants were midlife employees at baseline, and we focused on their premature exit from the workforce during the follow-up. We did not have information on other sleep or mental health disorders to use as criteria for exclusion in the surveys.

The ethical committee of the Faculty of Medicine, University of Helsinki has approved the study, and the City of Helsinki has provided permission to conduct this study. All participants could choose if they wanted to participate the surveys, and can at any time withdraw their consent.

### 2.2. Measures

#### 2.2.1. Indicator Variables of Sleep and Health Functioning for Repeated-Measures Latent Class Analysis (RMLCA)

##### Sleep Variables

The Jenkins Sleep Questionnaire (JSQ) was used to assess insomnia-related symptoms over the past 4 weeks [18]. The JSQ has been developed as a brief screening device with four items describing insomnia-related symptoms: (1) trouble falling asleep; (2) waking up several times per night without trouble falling asleep again; (3) waking up one or more times per night (including waking far too early) with trouble falling asleep again; and (4) waking up after usual amount of sleep feeling tired or worn out. The questionnaire has been validated mostly in clinical samples [19]. Answer options for all four questions were as follows: (1) not at all; (2) 1–3 nights; (3) 4–7 nights; (4) 8–14 nights; (5) 15–21 nights; (6) 22–28 nights. All variables were dichotomized in order to use them as indicator variables in Repeated Measures Latent Class Analysis (RMLCA) models (see Statistical analyses below). Options 1–3 were combined into a single category reflecting either not at all or only occasional sleep-related symptoms (variable value ‘1’) and options 4–6 were combined into a single category reflecting the existence of more or less severe sleep problems (variable value ‘2’). In addition, as an attempt to reduce the number of sleep indicators we calculated Spearman correlation coefficients between the four dichotomized items at baseline. The highest correlation (0.68) was found between items 2 and 3. When a separate latent class analysis was performed at baseline using items 2 and 3 as indicators, it was found that item 2 separated latent classes slightly better than item 3 (results not shown). Thus, items 2 and 3 were relatively strongly overlapping but item 2 was slightly more informative than item 3 for the purposes of this study. As a result, items 1, 2, and 4 were chosen for further analyses.

In addition, self-reported sleep duration at baseline was assessed by asking: How many hours on average do you sleep per night during the week? The response options were as follows: five hours or less, six hours, seven hours, eight hours, nine hours, and 10 h or more were classified into three groups: ≤6 (short sleepers), hours, 7–8 h (ref), and ≥9 h (long sleepers). A preliminary latent class analysis at baseline showed that sleep duration did not separate latent classes with an acceptable clarity (results not shown). Therefore, and as sleep duration did not add new information, it was not considered as an indicator variable in RMLCA models (see statistical analysis below) but instead as a potential characteristic of revealed RMLCA groups. Additionally, if it were added in the model, this would have made the models a lot more complex, because measuring sleep duration requires more categories. Using three categories for sleep duration would increase the number of possible response patterns from current 64 (2^3^ × 2^3^) to 216 (2^3^ × 3^3^).

##### Health Related Functioning

The SF-36 was used to measure physical and mental health functioning [10,20]. We used the Finnish translation [21]. The SF-36 measures health related functioning and wellbeing in eight subscales: (1) physical functioning (PF); (2) role limitations because of physical health problems (RP); (3) bodily pain (BP); (4) general health perceptions (GH); (5) general mental health (GM); (6) role limitations because of emotional problems (RE); (7) social functioning (SF); and (8) vitality (VI). GM, RE, SF, and VI are included in the mental domain of health and PF, RP, BP, and GH in the physical domain of health [22]. Two weighted aggregations of all eight subscales (summary scores) have been developed: (1) Physical Component Summary (PCS) and (2) Mental Component Summary (MCS). PCS and MCS capture more than 80% of the reliable variance in the eight subscales and have been introduced as an attempt to simplify the analysis and interpretation of the SF-36 [23] However, it has been argued that MCS and PCS may incorrectly summarize the information they are meant to represent [24] see also [23,25]. Therefore, we decided to use only SF-36 subscales in our analysis as indicator variables in competing RMLCA models to avoid misleading or inaccurate conclusions, and to produce a better understanding about the patterns of change in sleep and functioning and to show more accurate risk groups. However, we also initially performed RMLCA models with PCS and MCS as a sensitivity analysis. 

As an attempt to reduce the number of indicator variables we at first dropped off the subscale 8 (VI) because it attempts to measure tiredness and fatigue in the same way as does the JSQ item 4. Second, when the seven other SF-36 subscales were used as indicator variables in a latent class analysis at baseline it was found that the best latent class separation was obtained by using subscales BP, GH, GM, and SF (results not shown). Therefore, we considered only these four subscales in our further analyses as indicator variables to model normal and poor health functioning in RMLCA models. The lowest quartile was used as a cut point to dichotomize the four subscales for the purposes of the latent class analyses. Each original subscale included two to 10 items, and those included in the current models, two to five. Because of the varying number of items, standardized scores were used, as advised [20]. Higher scores indicate better functioning. Each transformed score has values between 0 and 100, and a value >75.0 indicated good functioning, and values lower than that indicated poor functioning. The distributions of the subscales are not normally distributed, and the number of different scores varies between the subscales.

#### 2.2.2. Other Variables

##### Outcome Variable

Premature exit from the labor market was the outcome variable. In Finland during 2000–2012 the general retirement age was flexible between 64 and 68 years. Consequently, only retirements before 64 years of age were considered as potential events, i.e., reflecting premature exit from paid employment. In some professions (e.g., teachers) the minimum retirement age is lower than 64 years. Therefore, only events such as taking an early retirement pension, unemployment pathway to retirement, or disability pension were combined into the outcome variable of premature exit from the labor market.

##### Background Variables

The baseline questionnaire included information on several pertinent potential background confounding factors [26]. Similar covariates have been used in numerous studies focusing on sleep, functioning, and work disability/premature exit from work due to disability (please see, e.g., refs. [4,5,6,11,14]). Briefly, the following sociodemographic background characteristics were assessed: age; gender; marital status (married or cohabiting, single, and, divorced/separated or widowed); education in three categories of basic, secondary, and higher education; occupational class was obtained through the employer’s personnel register in categories of managers or professionals, semi-professionals, routine non-manual employees, and manual workers. Managers have subordinates and do managerial/administrative work. They typically have a university degree. Professionals, including other upper white-collar employees, such as teachers and doctors, also have a university degree, but do professional work and they typically do not have subordinates. Semi-professional jobs require a college-level qualification or they can include supervisory but also routine tasks, which are characterized by having less autonomy. Semi-professionals in the municipal sector include, e.g., nurses, foremen and technicians, and other intermediate level white-collar employees. Routine non-manuals and manual positions require vocational training or they have no specific qualifications, with routine-non manuals being employed in non-supervisory clerical or non-manual tasks, and manual workers in e.g., transportation or cleaning. The following health-related background characteristics were assessed: current smoking (yes/no); frequency of drinking alcoholic beverages (in categories of not at all, max once per month, 2–4 times per month, and 2–7 times per week); the last 12 months’ leisure-time physical activity or commuting were asked using 4 grades of intensity (from 0 to 4 h or more) for walking, brisk walking, jogging, and running or their equivalent activities [27]. Metabolic equivalents (METs) were used to approximate the amount of physical activity. MET hours per week were calculated by multiplying the time spent in physical activity with the MET value of each intensity grade and adding the 4 values together. The variable was dichotomized in a way that participants with weekly exercise < 14 MET hours per week were classified as physically inactive. Body mass index (BMI) was calculated as a continuous variable based on participants’ self-reported weight and height. Limiting long-standing illness was assessed as follows: Do you have any long-standing illness, disability, or infirmity? (Yes/no). If the answer to the question of long-standing illness was yes a follow-up question was asked: Does your illness/disability restrict your work or does it limit your daily activities (gainful employment, housework, schooling, studying)? Those who answered yes were classified as having a limiting long-standing illness.

In addition, intention to retire early was assessed with: Have you considered retiring before normal retirement age? Possible answers were as follows: (1) No, I have not; (2) Yes, sometimes; (3) Yes, often; and (4) I have already submitted a pension application. Self-assessment of the opportunities to continue working until the official retirement age was assessed with: Do you think you will be able to continue working until your normal retirement age for your job?: (1) most likely; (2) not sure; and (3) I do not think so.

### 2.3. Statistical Analysis

Premature exit from work is a classic example of time-to-event count data. Time-to-event analyses are not possible within PROC LCA (SAS v9.4). Therefore, the classify-analyze approach with 2 steps was chosen. First, classification of individuals using a person-oriented approach by repeated measures latent class analyses (RMLCA). Second, time-to-event analysis of premature retirements between defined RMLCA groups by Cox’s proportional hazards modelling (hazard ratios, HR and their 95% confidence intervals).

#### 2.3.1. First Analytical Step: RMLCA Based Classification

At the first analytical step, RMLCA, was used to fit models of change in sleep and health functioning over the three phases of data collection. The approach is similar to growth mixture modelling in the sense that each latent class is associated with a characteristic vector of responses over time [28]. However, in RMLCA no functional form of the growth curve is fitted. Change in sleep and health functioning was modelled in whatever form they naturally occur in each latent class [28]. Because RMLCA works best with only a small number of indicator variables [28] we decided to use two dichotomized indicators (one for sleep and one for health functioning) in three study phases yielding a contingency table of 2^3^ × 2^3^ = 64 cells as a starting point for each RMLCA model. Thus we had 64 potential response-patterns or ’trajectories’. All these response patterns form a matrix or array of response patterns. If we added, for example, even one additional indicator in our model, the number of possible response patterns would increase to 512 (2^3^ × 2^3^ × 2^3^), and adding a fourth one, the number would be 4096 (2^3^ × 2^3^ × 2^3^ × 2^3^), etc. Including 7 indicators into the model at the same time would mean that we would end up with more than 2 million possible response patterns to analyze (7 dichotomous variables in 3 time points, i.e., 2^3^ × 2^3^ × 2^3^ × 2^3^ × 2^3^ × 2^3^ × 2^3^= 2,097,152). If the indicator variables had more than two categories, the numbers would also be notably higher. As pointed out in the measurements section (see above) there were three dichotomized indicators of insomnia-related symptoms (JSQ items 1, 2 and 4) and four dichotomized indicators of health functioning from SF-36 (subscales BP, GH, GM, and SF). Consequently, altogether there were 3 × 4 = 12 possible RMLCA models. These can be seen as 12 different operationalizations of one model where the two underlying constructs are insomnia and health functioning. For each of these models, a series of consecutive solutions with different numbers of latent classes (from one to seven) were estimated with SAS PROC LCA (v9.4). Individuals were assigned to the best-fitting class based on their probability of class membership. Only those solutions where homogeneity and latent class separation were considered to be sufficiently good enough for reliable classification were considered in this analytical step.

#### 2.3.2. Second Analytical Step: Age-at-Event Analysis

We aimed to analyze whether identified RMLCA groups (based on characteristic changes in sleep and health functioning over time) were associated with the risk of premature exit from work. Therefore, the second analytical step was a time-to-event analysis of premature retirements between the identified RMLCA groups. That was performed by Cox proportional hazards modelling. Because a person’s age is an obvious determinant of retirement, age was used as the time-scale in the Cox models instead of time-on-study [29]. Because of that we call this an age-at-event analysis. We conducted both unadjusted and adjusted hazard ratios (HR) models. The adjusted models included adjustment for gender, education, physical activity, smoking, alcohol, limiting longstanding illness, BMI, and socioeconomic status.

## 3. Results

### 3.1. Descriptive Characteristics of the Study Sample

Baseline background characteristics of the study participants are shown in Table 1. The mean age was 55.5 years. A majority of women and men were married, and roughly a third had higher education. Also about a third reported a limiting long-standing illness. At baseline, 50% had not yet considered premature retirement (before their actual retirement age), while 15% had often considered it.

Table 2 describes the insomnia-related symptoms, mental health, physical, and social functioning (used as indicator variables in RMLCA models) across the study phases in the people in this study.

### 3.2. First Analytical Step: RMLCA Based Classification

RMLCA Models of Change in Sleep and Health Functioning Over Three Phases of the Study.

Altogether 12 possible combinations of two (one sleep and one health functioning) indicator variables (four indicators of functioning × three indicators of sleep) were analyzed. In other words, we tried to define different patterns of movements in and out of problem categories of sleep and functioning across study phases. For each model, a series of different solutions (1–7 latent classes) was estimated and the best of them was selected considering the balance between model’s absolute fitness (minimizing Bayesian Information Criteria, BIC) and parsimony, latent classes’ homogeneity (reliability of posterior class membership probability > 0.700), and their interpretation.

The best solution was always considered to be a four-class solution. In Table 3 all 12 models with four-class solutions are presented. The table also shows whether the model was accepted or rejected. 

In our additional sensitivity analyses using PCS and MCS in place of the subscales (6 models/figures), the broad picture of the results were very similar. Thus, we identified patterns of change in each model that can be described as subgroups of having consistently poor functioning and poor sleep, consistently good functioning and sleep, poor functioning but good sleep, and good functioning but poor sleep. For specific insomnia symptoms, there was an indication of different levels of problems, but the patterns of change could be classified similarly. Due to the problems described in the methods section, and due to our aim to increase our understanding about the more specific risk groups and patterns of change in sleep and functioning, we preferred to retain the subscales in our main models.

Importantly, across different models the interpretation of defined latent classes in acceptable models were relatively similar. Latent Class 1 (LC1) was labelled as the reference class (ref.) and was characterized by the absence of any kind of sleep problems throughout all study phases. Latent class 2 (LC2) was named as persistent sleep problems. The class was characterized by high probability of sleep problems (sometimes with increasing trend) with absence (or low probability) of poor health functioning. Latent class 3 (LC3) was named as poor health functioning and was characterized by stable (or increasingly) high probability of poor health functioning with absence or low probability of sleep problems. Latent class 4 (LC4) was named as problematic sleep and health functioning and was characterized with stable (or increasingly) high probability of poor functioning both in sleep and health indicators. 

The prevalence of membership in the latent classes in different acceptable RMLCA models varied as follows: LC1 (48.0–73.0%); LC2 (5.2–17.9%); LC3 (10.8–28.2%); and LC4 (5.0–14.6%). Illustrations of the latent class profiles for interpretation are shown in Figure 1a–e.

### 3.3. Second Analytical Step

Age-at-Event Analysis for Premature Retirement Events between Different RMLCA Groups. Across the study phases 2065 retirements took place. Of these 1702 were statutory old age retirements and 363 were caused by other reasons, mainly disability pension events. These 363 retirements were considered as adverse outcome events in age-at-event analyses using Cox’s proportional hazards modelling to compare the risk of premature retirement events in different latent classes in the RMLCA model. Individuals were assigned to groups indicated by the model’s latent classes using the maximum probability assignment rule. Each individual was assigned to the group indicated by maximum posterior probability. The reliability of this classification is shown in Table 3. This reliability refers to the latent class model where the classification error is taken into account and each individual has a membership probability (different from zero) in each latent class of the model. When the real classification is conducted, each individual belongs to the group where their posterior probability was highest and it is now set to be 1. Membership probability in all other groups is then 0. Therefore, classification error cannot be taken into account anymore and prevalence figures of the groups somewhat differ from prevalence figures of latent classes.

The results of these age-at-event models was that all models were statistically significantly associated with unwanted retirements events. The strength of the association depended on RMLCA model indicators, type of the RMLCA class and number of latent classes in the model. The strongest associations were found by models with indicators of SF-36 subscale GH with difficulties in initiating sleep or nonrestorative sleep. Additionally, strong associations were found in a model with SF-36 subscale BP and difficulties in initiating sleep, as well as in the model with SF-36 subscale SF and difficulties in initiating sleep. Interestingly, models with indicators of mental functioning (GMH subscale) revealed the weakest (although still statistically significant) associations with difficulties in initiating sleep. The strongest risk (independent of model indicators) was always found among members of latent class 4 (Problematic sleep and health functioning), the group with a relatively stable and high level of problems in both health functioning and sleep. The fully adjusted HRs for premature exit from work in this group varied between 2.19 and 5.17 when compared with the reference group. The next strongest risk was found among members of latent class 3 (Poor health functioning), the group with prevalent problems in different dimensions of health functioning which either were at a stable level or were increasing during the study phases but almost without sleep problems. The fully adjusted HRs for premature exit from work in this group varied between 1.64 and 3.20 when compared with the reference group. Class 2 (persistent sleep problems), the group with high probability of sleep problems which were either relatively stable or increasing during study phases but who were characterized by only a low probability of health functioning problems were next. HR estimates in this group varied between 1.18 and 2.96. The above-mentioned Cox models with the lowest and highest HR estimates of each latent class are shown in Table 4. In addition, Figure 2a–c depict as examples unadjusted Cox proportional hazard models. LC groups of RMLCA models with the highest HR for each latent class are shown, to display in which model and with which indicators each of the LC groups had the strongest association in terms of the HR.

### 3.4. Baseline Self-Reported Sleep Duration Differences between RMLCA Groups

The RMLCA groups were built using insomnia-related symptoms and SF-36 health functioning subscales as latent class indicators. Their predictive power remained even after adjustments for several pertinent covariates. However, because of obvious over adjustment, self-reported sleep duration was not included among covariates. Therefore, we analyzed baseline self-reported sleep duration distributions across RMLCA groups. Prevalence of baseline sleep duration groups are shown in Table 5 by RMLCA groups with their highest and lowest fully adjusted HR estimates.

There was a crude positive association between the predictive power of RMLCA groups and the prevalence of short sleepers at baseline. The highest prevalence of short sleepers was found among the group with Problematic sleep and health functioning (LC4), which was the most powerful estimator of premature exit from work. The prevalence of short sleepers (>40%) in that group was about 2.2 times higher than in the reference group (in all eight models *p* < 0.0001) and it was also always higher than in the group with LC3 poor health functioning (*p* < 0.0001). With one exception it was also always higher than in LC2 persistent sleep problems (*p* < 0.010). The prevalence of short sleepers was lowest in the reference group, which differed significantly from all other groups except in four models from LC3 poor health functioning (*p* < 0.400). In addition, short sleep duration differed less consistently between LC2 and LC3 (in all eight models *p* < 0.06). Long sleep duration did not generally differ between RMLCA groups. In only two cases, among all accepted model solutions, there was a statistically significant difference found in the proportion of long sleepers.

### 3.5. Third Analytical Step: Conventional Variable Oriented Approach

In a conventional variable-oriented approach the aim of the analyses would be to test whether sleep and health functioning variables (which were used as indicator variables in RMLCA) were associated with the outcome in a statistically independent way. The fully adjusted Cox models with baseline SF-36 subscales and insomnia-related Jenkins items as risk factors for premature retirement suggested that in most cases they were. However, the strength of the association was weaker than that found in person-oriented approach between RMLCA groups and the outcome. Variable oriented analyses indicated that HR estimates for SF-36 subscales varied between 1.04 and 2.04. Statistically non-significant estimates were also found in models with the GMH subscale. The HR estimates for Jenkins insomnia-related variables varied between 1.52 and 1.95. The interaction term (SF-36 subscale by sleep) was always statistically non-significant. When sleep duration was added into the fully adjusted models it was found that in all models the effect of short sleep was always statistically non-significant (HR varied between 1.06 and 1.10, *p* always >0.54). Conversely, the association with long sleep was always statistically significant (HR varied between 1.88 and 2.05, *p* always <0.008).

## 4. Discussion

### 4.1. Main Findings

This study aimed to first identify groups of employees characterized by their patterns of change in insomnia-related symptoms and health functioning across a 10-year follow-up period. Then, we examined the associations between the identified groups and premature exit from work. Using different combinations of sleep difficulties and health functioning dimensions in 12 RMLCA models we identified four homogenous subgroups of employees with characteristic developmental vectors of changes in sleep and health functioning. Although the prevalence of defined subgroups varied between different RMLCA models and model solutions, the interpretation of them across different models was relatively consistent revealing comparatively clearly defined subgroups of employees with differences in their risk of premature exit from work.

### 4.2. Interpretation

As expected, the lowest risk was found among a subgroup characterized by good sleep and health functioning over the whole follow-up period. The group was also most prevalent. Irrespective of RMLCA model the highest risk of premature exit from work was always found among the members of a subgroup characterized by stable or increasingly high probability for both poor sleep and health functioning (Problematic sleep and health functioning, LC4). The next highest risk was found in a subgroup characterized by stable or increasingly high probability of poor health functioning with concurrently absent or low probability of sleep problems (Poor health functioning, LC3). On average a somewhat lower, although not statistically significantly, risk was found among a subgroup characterized by constantly high or increasing probability of sleep problems with concurrently good health functioning (persistent sleep problems, LC2).

To the best of our knowledge, previous studies of premature/disability retirement have all used variable-oriented approaches analyzing relations between variables and assuming that these relations apply across all individuals [28]. Instead, we used RMLCA as a person-oriented approach looking for subtypes of individuals that exhibit similar patterns of these risk factors [28]. In these models the groups thus arise from the data, and they are not pre-defined by the researcher as in the variable-oriented models. Our additional variable-oriented analyses helps highlight the value of the person-oriented models. Thus, we may miss important risk groups, if we merely use independent variables as predictors of the outcome, as such analyses assume that the associations apply to all people. This could be less precise than a RMLCA, as a variable-oriented approach misses change over time and also only shows average associations. In addition, we studied directly the intended final composite outcome (premature exit from the labor market, including taking early retirement pension, unemployment pathway to retirement, or disability pension). Several previous studies have analyzed preceding surrogate outcomes like sickness absence or intention to retire early. 

Sickness absence as a generic indicator of overall health and work-related disability has been shown to be associated with future health outcomes [30,31]. Although mental disorders are well known to increase the risk of sickness absence [9] it has been reported in a previous analysis of the Helsinki City study that SF-36 subscales measuring physical domains of functioning, especially pain and general health perception, are stronger predictors of sickness absence than the mental subscales [11]. The results from our separate variable-oriented analysis concerning premature retirement supports this. The subscale of general mental health was not independently associated with premature retirement when insomnia-related symptoms were included into the Cox model. However, pain and general health perception were. But, when the RMLCA groups were used as the independent variable in the Cox models, it was found that LC4 always, irrespective of its indicators, had the strongest associations with premature retirement. When LC4 was defined using SF-36 general health perception subscale as an indicator it had its strongest statistically predictive power. The second strongest predictive power was suggested when using bodily pain, followed by social functioning. The general mental health subscale had the lowest, but still statistically significant predictive power, when it was used in defining LC4. Consequently, in variable-oriented analyses, as expected, the effect of general mental health is accounted for by insomnia-related factors [32,33]. Our person-oriented approach indicated that the decisive factor for the effect of mental health problems is their chronicity, indicated by their stability over the entire follow-up in repeated measurements and the associations with early exit from the workforce for this developmental group.

Insomnia-related symptoms have been repeatedly shown to be associated with disability retirement [33] and sickness absence [3,34,35] including previously in these HHS data [36]. Furthermore, the joint contributions of insomnia and pain to the risk of disability retirement have been stronger than the individual effects of insomnia [37]. In line with this, our person-oriented analysis indicated that when examining the associations for premature retirement by classifying individuals into subgroups, insomnia-related symptoms and general health perception or bodily pain yielded the most predictive subgroups of individuals with increased likelihood of early exit. In addition, our results emphasize the importance of the stability of insomnia-related symptoms which increased the risk of premature exit from work among members of LC4 and LC2. The role of sleep problems may be partly explained by reciprocal associations between work characteristics and sleep problems. Some work characteristics influence future sleep and sleep may affect future psychosocial risks at work [38]. We defined the RMLCA subgroups using different combinations of three insomnia-related symptoms and four SF-36 subscales. It has been previously shown that insomnia symptoms are associated with poor (lower) scores in all eight dimensions of the SF-36 subscales [39]. However, our person-oriented analysis showed that in time they have developmental trajectories which are not always tied to each other. There are subgroups of individuals in whom these trajectories differ markedly like in LC2 and LC3. Importantly, this had an effect on the risk of premature exit from work among these groups suggesting that sleep and health functioning are both independent risk factors.

Previous studies have suggested that insomnia-related symptoms have the strongest effect on sickness absence when insomnia symptoms are coexisting with short sleep duration [40]. Extreme ends of the sleep duration distribution have also been associated with future sickness absence [41]. In our separate variable-oriented analysis we found that in all Cox models the effect of short sleep (≤6 h) was always accounted for by insomnia-related symptoms, but the effect of long sleep (≥9 h) was always statistically significant. However, the RMLCA subgroup with the highest risk for premature exit from work (LC4) always had a clearly higher proportion of short sleepers than any other subgroup. Consequently, self-reported short sleep duration might be an important factor indicating increased risk among individuals with concurrent insomnia-related symptoms and health functioning problems. The prevalence of long sleepers did not generally differ among RMLCA groups indicating that, although being a risk factor, it did not account for the risk differences between RMLCA groups. This result may be related with the finding that self-reported estimates of habitual sleep duration are reported to be imprecise and the magnitude of the bias is related to other key health determinants [42].

Future studies could aim at addressing the changes in sleep and functioning across more phases all preceding the retirement event, or focus also on the patterns of changes after retirement. Another approach could be to examine joint development of sleep and functioning by fitting dual trajectory models with continuous variables, and using the group membership as a determinant of later retirement events.

### 4.3. Methodological Considerations

#### 4.3.1. Limitations

Our study has limitations. First, as this study is observational, no causal inferences can be made. Thus, we show associations between the identified latent groups and age at exit from paid employment during the follow-up. More specifically, our design using RMLCA to identify patterns of change in sleep and functioning over 10 to 12 years, and examining premature exit over the three time points, does not allow for causal predictions to be made. There was a relatively large attrition among participants who answered all three questionnaires (1660 individuals 24.4%) resulting in missing information on some sleep or SF-36 variables, i.e., on the indicator variables of the RMLCA models. If this attrition is not completely random it may have influenced the composition of RMLCA subgroups. However, we had seven indicators among which we analyzed 12 possible RMLCA models and found four-class solutions to be the best models with relatively consistent interpretation across all models. If the attrition were selective picking out one or few indicators but not others it would have produced some deviant (not sharing the general interpretation) latent classes. Because that was not found, we believe that the attrition did not have meaningful effects on our models. In addition, we analyzed one of the models including all individuals (i.e., also those with missing information on some indicator variable/s in some study phase). We found only very slight differences in the class prevalence (0.1% to 0.7%), and classification error very slightly increased (e.g., reliability of LC2 was decreased from 0.840 to 0.834) without any practical meaning when compared with the model solution with individuals with complete information. Consequently, we think our results are statistically reliable. We acknowledge that there are different ways to handle missing data, such as maximum likelihood estimation. None of the techniques for missing data is without limitations, as compared to actually having complete data (no missing responses). As these data appeared to be largely missing at random, any technique to handle missing data is, however, unlikely to notably change the results. Still, we acknowledge that due to missing data, we are left with a degree of uncertainty. We should also consider generalizability of these findings. This cohort was female-dominated and represented only the municipal sector and their largely female midlife employees. Any further generalization to other employment sectors, countries or different age groups should be done with extreme caution. Finally, it needs to be noted that there is no ‘absolute’ test to decide the best indicators but the decisions by necessity have to be made by the researchers [43]. Finally, it was not possible to conduct analyses stratified by season, as most surveys were collected in autumn. Further research could aim to establish, if the associations are affected by the four seasons and seasonal effects in the reports of insomnia.

#### 4.3.2. Strengths

This study also had some strengths. First, we had a large sample of midlife employees, who were all employed at baseline. This enabled us to follow them up for their risk of early exit from workforce during the 10 to 12 year follow-up. The same individuals responded to the surveys three times, and survey items at each time point were identical. Thus, we could apply the RMLCA and study the developmental patterns of change in sleep and functioning, and how they are associated with early exit. The measures of insomnia symptoms and health functioning were validated and have been used in numerous studies for decades [18,21]. As such models have been seldom applied, we provided a detailed description of our models and in addition to providing the results, we also provide this as an example and a tutorial for other researchers in the field to apply in their research questions, potentially producing new information about latent risk groups and their determinants.

## 5. Conclusions

Person-oriented repeated measures latent class analyses (RMLCA) can be used to compliment traditional variable-oriented methods, and pinpoint risk groups that were previously missed. In the present analyses looking at premature exit from the workforce of municipal employees of the City of Helsinki we found that the associations gleaned from this method may provide both qualitatively different and quantitatively more comprehensive view and show better predictive strength to conventional risk factor methodology. Workers at the City of Helsinki are likely at heightened risk of early workforce exit particularly when they have poor or worsening sleep and/or health functioning. It would be very useful if this analysis could be repeated in a study with more than three time points.

## Figures and Tables

**Figure 1 ijerph-18-01725-f001:**
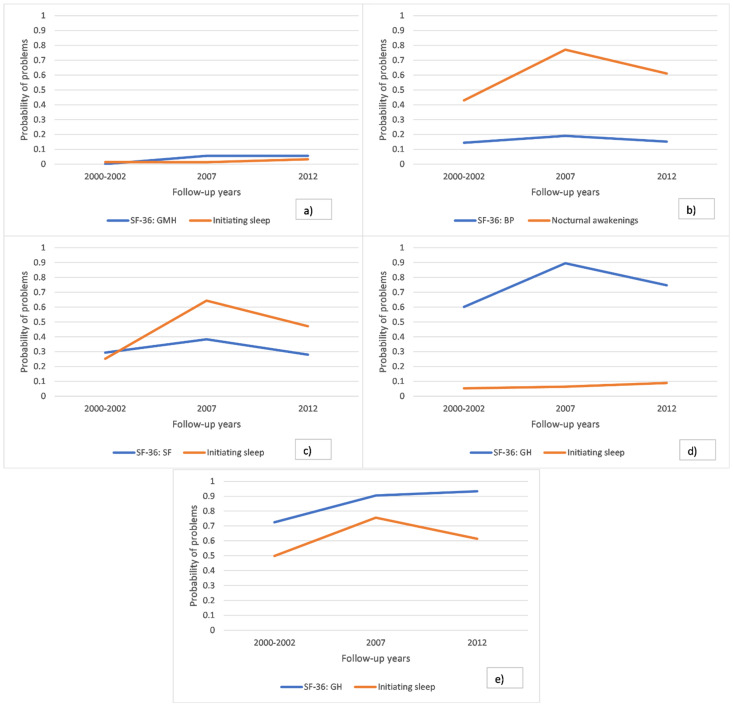
(**a**) An example of LC1 (reference class: stable good sleep and good functioning), from the RMLCA model with General Mental Health and Difficulties of initiating sleep as indicators of the latent classes. (**b**) An example of LC2 (persistent sleep problems), from the RMLCA with Bodily Pain and Nocturnal awakenings as indicators of the latent classes. (**c**) Another example of LC2 (persistent sleep problems), from the RMLCA with Social Functioning and Difficulties of initiating sleep as indicators of the latent classes. (**d**) An example of LC3 (Poor health functioning), from the RMLCA with General Health perception and Difficulties of initiating sleep as indicators of the latent classes. (**e**) An example of LC4 (Problematic sleep and health functioning), from the RMLCA with General Health perception and Difficulties of initiating sleep as indicators of the latent classes.

**Figure 2 ijerph-18-01725-f002:**
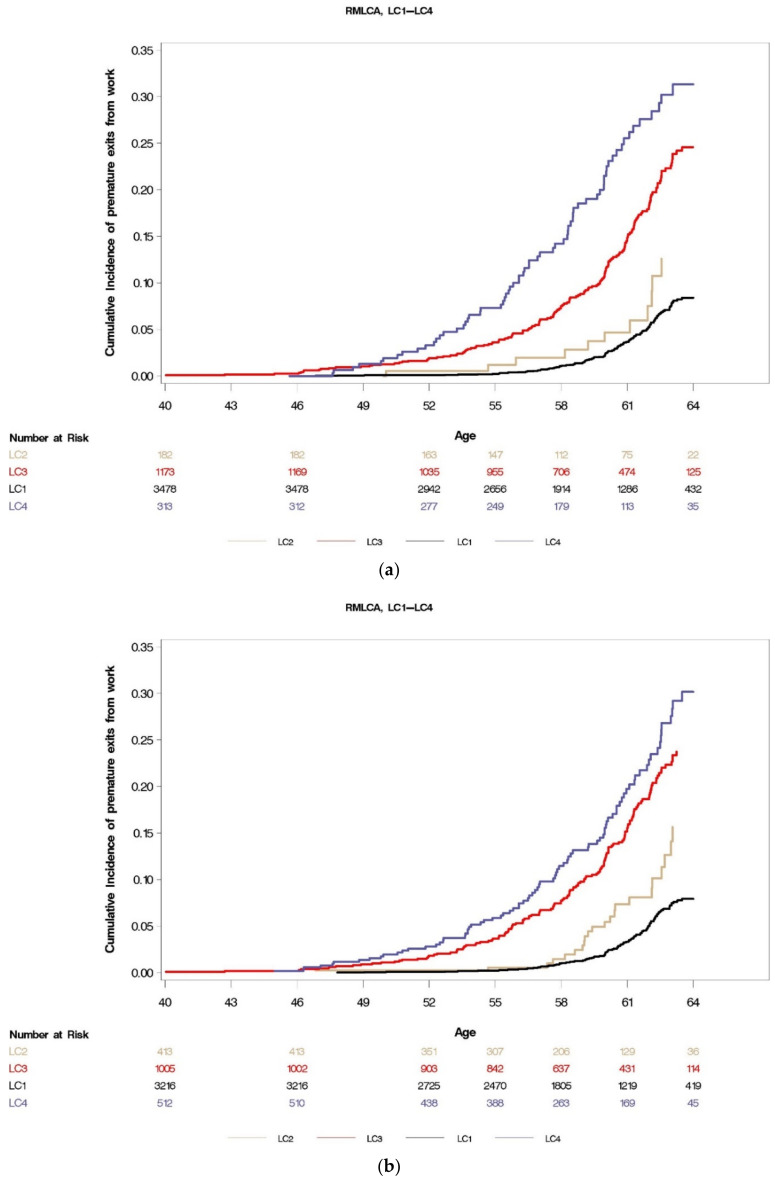
(**a**) RMLCA model of General Health Perception and Problems of Initiating Sleep (LC1 as a reference). In this model the predictive power of LC4 is at its maximum (HR = 6.48). (**b**) RMLCA model of General Health Perception and Nonrestorative Sleep (LC1 as a reference). In this model the predictive power of LC3 is at its maximum (HR = 4.21). (**c**) RMLCA model of Social functioning and Nonrestorative Sleep (LC1 as a reference). In this model the predictive power of LC2 is at its maximum (HR = 3.14).

**Table 1 ijerph-18-01725-t001:** Baseline background characteristics of the study participants.

	All (*n* = 5148)	Women (*n* = 4204)	Men (*n* = 944)
Sociodemographic characteristics			
Gender (%)		82	18
Age (mean ± SD)	55.5 ± 6.6	55.2 ± 6.5	56.7 ± 6.5
Marital status (%)			
Married or co-habiting	70.7	68.6	79.9
Single	12.6	13.2	9.8
Separated or widowed	16.7	18.2	10.3
Education (%)			
Basic	38.2	39.0	34.7
Secondary	33.9	35.1	28.6
Higher	27.9	25.9	36.6
Occupational class (%)			
Managers & professionals	31.9	28.3	47.9
Semi-professionals	21.4	21.7	20.4
Routine non-manual	33.0	38.3	9.2
Manual workers	13.7	11.7	22.4
Health behavior			
Leisure-time or commuting physical activity (%)			
Physically inactive (MET < 14)	23.2	23.4	22.6
Physically active (MET ≥ 14)	76.8	76.6	77.4
Current smoking (%)			
Yes (%)	21.3	21.0	22.9
Alcohol (%)			
Not at all	6.9	7.2	5.4
max once/month	32.3	35.0	20.3
2–4 times/month	37.0	36.9	37.3
2–7 times/week	23.9	21.0	37.0
Health status			
Limiting longstanding illness (%)	30.5	30.8	29.4
BMI (mean ± SD)	25.4 ± 4.3	25.2 ± 4.4	26.3 ± 3.7
Considered retiring before your actual retirement age (%)			
No, I have not	50.4	51.4	46.2
Sometimes	33.2	33.0	34.5
Often	15.1	14.4	17.9
I have already submitted an application	1.3	1.2	1.4
Estimated ability to continue at your work until your normal retirement age (%)			
Most likely	57.5	56.0	63.9
Not sure	34.7	36.3	27.8
I do not think so	7.8	7.7	8.3

**Table 2 ijerph-18-01725-t002:** Descriptive characteristics of sleep and health functioning in the study participants across study phases. Values are shown as percent of problems indicated by dichotomized indicator variables for repeated measures latent class analyses (RMLCA) models.

	All (*n* = 5148)			Women (*n* = 4204)			Men (*n* = 944)		
	Phase1	Phase2	Phase3	Phase1	Phase2	Phase3	Phase1	Phase2	Phase3
	%	%	%	%	%	%	%	%	%
Jenkins insomnia-related symptoms									
Difficulties falling asleep:	7.4	11.1	10.9	7.7	11.5	11.3	6.0	9.0	9.2
Nocturnal awakenings ^1^	22.4	30.4	29.9	22.8	31.1	30.3	20.2	27.3	28.2
Non-restorative sleep ^2^	20.5	22.0	19.0	21.2	23.0	19.7	17.7	17.3	15.7
SF-36 Health functioning subscales									
SF-36 BP: poor	26.8	30.3	30.8	25.2	28.9	29.4	33.6	36.6	37.3
SF-36 GH: poor	25.6	32.0	33.2	25.5	32.5	34.1	25.9	29.9	29.0
SF-36 GMH: poor	18.7	18.8	17.1	17.5	17.7	16.2	24.5	23.7	21.2
SF-36 SF: poor	33.9	35.0	33.7	32.2	33.6	32.1	41.4	41.0	40.7

^1^ JSQ item 2: wake up several times per night without trouble falling asleep again. ^2^ JSQ item 4: wake up after usual amount of sleep feeling tired or worn out. SF-36: BP = Bodily pain subscale; GH = General health perceptions; GMH = General Mental Health subscale; SF = Social Functioning subscale.

**Table 3 ijerph-18-01725-t003:** RMLCA models with four class solutions. Whether the model was chosen or rejected is indicated. Membership prevalence (%) and reliability (rel.) of posterior class membership probability (between 0 and 1.0) are also shown for each latent class.

Indicators of Latent Classes	G ^2^	BIC	Proportion of Individual’s in Each of the Latent Classes (%) and Mean Classification Probability in Each Class				Chosen/Rejected
			LC1% rel.	LC2% rel.	LC3% rel.	LC4% rel.	
Bodily pain (BP) + Difficulties of initiating sleep	86.4	317.1	62.0%	6.4%	25.5%	6.2%	chosen
			0.890	0.724	0.840	0.835	
Bodily pain (BP) + Nocturnal awakenings	164.2	395.0	49.1%	17.9%	20.5%	12.6%	chosen
			0.870	0.725	0.774	0.827	
Bodily pain (BP) + Non-restorative sleep	195.8	426.5	55.0%	11.7%	23.3%	10.0%	chosen
			0.873	0.799	0.755	0.820	
General health perception (GH) + Difficulties of initiating sleep	75.4	306.2	63.9%	5.2%	24.3%	6.7%	chosen
			0.920	0.763	0.861	0.818	
General health perception (GH) + Nocturnal awakenings	410.1	640.9	53.0%	8.0%	27.5%	11.4%	rejected ^1^
			0.957	0.873	0.891	0.909	
General health perception (GH) + Nonrestorative sleep	212.1	442.9	58.3%	10.9%	19.9%	11.1%	chosen
			0.900	0.801	0.785	0.829	
General mental health (GMH) + Difficulties of initiating sleep	133.0	363.6	73.0% 0.925	7.0% 0.743	15.0% 0.821	5.0% 0.858	chosen
General mental health (GMH) + Nocturnal awakenings	212.2	443.0	55.6%	12.4%	16.6%	15.4%	rejected ^2^
			0.905	0.864	0.627	0.839	
General mental health (GMH) + Nonrestorative sleep	247.7	478.4	67.9%	11.4%	10.8%	9.9%	
			0.921	0.762	0.777	0.912	chosen
Social functioning (SF) + Difficulties of initiating sleep	143.7	373.6	58.6% 0.892	7.1% 0.762	28.2% 0.810	6.2% 0.752	chosen
Social functioning (SF) + Nocturnal awakenings	183.6	414.3	48.0%	17.3%	23.0%	11.8%	chosen
			0.857	0.770	0.800	0.757	
Social functioning (SF) + Nonrestorative sleep	254	485	56.4%	8.7%	20.3%	14.6%	
			0.904	0.766	0.731	0.876	chosen

^1^ The model was rejected because of interpretational difficulties. Thus, there was no meaningful interpretation for the latent classes, or how the classes were distinct with respect to the phenomenon examined. ^2^ The model was rejected because at least in one latent class the mean classification probability (reliability) was <0.700. LC1 = Reference group; LC2 = Persistent sleep problems; LC3 = Poor health functioning; LC4 = Problematic sleep and health functioning.

**Table 4 ijerph-18-01725-t004:** Selected Cox models using RMLCA model based classification as a statistical predictor of premature retirement events (*n* = 363 events, ref.= reference group, latent class 1, LC1).

SF-36: GH + Difficulties in initiating sleep as indicators of RMLC: 4 class solution (See also Figure 2a)					
Bivariable Cox model ^1^			Adjusted Cox model ^2^		
df = 3; Wald χ^2^ = 190.0; *p* < 0.0001			df = 3; Wald χ^2^ = 103.1; *p* < 0.0001		
LC	HR	95% CL	LC	HR	95% CL
LC1	ref	-	LC1	ref	-
LC2	1.69	0.91–3.13	LC2	1.45	0.73–2.88
LC3	3.96	3.12–5.03	LC3	3.10	2.35–4.10
LC4	6.48	4.78–8.78	LC4	5.17	3.68–7.28
SF-36: GMH + Nonrestorative sleep as indicators of RMLC: 4 class solution					
Bivariable Cox model ^1^			Adjusted Cox model ^2^		
df = 3; Wald χ^2^ = 64.4; *p* < 0.0001			df = 3; Wald χ^2^ = 43.4; *p* < 0.0001		
LC	HR	95% CL	LC	HR	95% CL
LC1	ref	-	LC1	ref	-
LC2	2.42	1.78–3.29	LC2	2.27	1.62–3.17
LC3	2.26	1.67–3.05	LC3	2.07	1.50–2.87
LC4	2.41	1.78–3.25	LC4	2.19	1.58–3.05
SF-36: GH + Nonrestorative sleep as indicators of RMLC: 4 class solution (See also Figure 2b)					
Bivariable Cox model ^1^			Adjusted Cox model ^2^		
df = 3; Wald χ^2^ = 177.7; *p* < 0.0001			df = 3; Wald χ^2^ = 99.3; *p* < 0.0001		
LC	HR	95% CL	LC	HR	95% CL
LC1	ref	-	LC1	ref	-
LC2	1.83	1.15–2.9	LC2	1.95	1.18–3.21
LC3	4.21	3.27–5.43	LC3	3.20	2.39–4.28
LC4	5.60	4.19–7.46	LC4	4.88	3.52–6.77
SF-36: GMH + difficulties in initiating sleep as indicators of RMLC: 4 class solution					
Bivariable Cox model ^1^			Adjusted Cox model ^2^		
df = 3; Wald χ^2^ = 65.4; *p* < 0.0001			df = 3; Wald χ^2^ = 37.2; *p* < 0.0001		
LC	HR	95% CL	LC	HR	95% CL
LC1	ref	-	LC1	ref	-
LC2	2.63	1.96–3.52	LC2	2.11	1.53–2.90
LC3	1.79	1.33–2.42	LC3	1.64	1.18–2.28
LC4	2.88	1.96–4.22	LC4	2.54	1.69–3.81
SF-36: SF + Nonrestorative sleep as indicators of RMLC: 4 class solution (See also Figure 2c)					
Bivariable model ^1^			Adjusted model ^2^		
df = 3; Wald χ^2^ = 111.6; *p* < 0.0001			df = 3; Wald χ^2^ = 70.1; *p* = < 0.0001		
LC	HR	95% CL	LC	HR	95% CL
LC1	ref	-	LC1	ref	-
LC2	3.14	2.22–4.43	LC2	2.96	2.04–4.31
LC3	2.89	2.21–3.78	LC3	2.48	1.84–3.33
LC4	3.78	2.87–4.98	LC4	3.27	2.40–4.46
SF-36: PB + Nocturnal awakenings as indicators of RMLC: 4 class solution					
Bivariable Cox model ^1^			Adjusted Cox model ^2^		
df = 3; Wald χ^2^ = 147.8; *p* < 0.0001			df = 3; Wald χ^2^ = 69.6; *p* < 0.0001		
LC	HR	95% CL	LC	HR	95% CL
LC1	ref	-	LC1	ref	-
LC2	1.23	0.87–1.74	LC2	1.18	0.81–1.71
LC3	2.93	2.21–3.88	LC3	2.06	1.50–2.82
LC4	4.68	3.56–6.14	LC4	3.41	2.51–4.62

^1^ Unadjusted. ^2^ Adjusted for gender, education, physical activity, smoking, alcohol, limiting longstanding illness, BMI and socioeconomic status. SF-36: BP = Bodily pain subscale; GH = General health perceptions; GMH = General Mental Health subscale; SF = Social Functioning subscale. LC1 = Reference group (ref.); LC2 = persistent seep problems; LC3 = Poor health functioning; LC4 = Problematic sleep and health functioning.

**Table 5 ijerph-18-01725-t005:** Distribution of baseline sleep duration across RMLCA groups by their strength of association.

RMLCA Group	Maximum and Minimum Predictive Power (HR) of a Given RMLCA Group in Fully Adjusted Cox Models Predicting Premature Exit from Work	Self-Reported Baseline Sleep Duration (%)		
		≤6 h	7–8 h	≥9 h
LC1 Reference	-	18.7 *	77.6 *	3.7 *
LC4 Problematic sleep	HR max 5.17	42.4	52.8	4.9
and health functioning	HR min 2.19	41.8	55.9	2.3
LC3 Poor health functioning	HR max 3.20	22.9	72.9	4.2
	HR min 1.64	25.4	70.8	3.8
LC2 Persistent sleep problems	HR max 2.96	30.2	66.5	3.3
	HR min 1.18	25.3	72.2	2.5

* Mean across eight models. HR = Hazard ratio.

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
