# Peer review of "Associations of Sleep and Health Functioning with Premature Exit from Work: A Cohort Study with a Methodological Emphasis"

_ijerph, 2021, doi:10.3390/ijerph18041725_

Round 1

Reviewer 1 Report

Thank you for the opportunity to review this interesting paper. My comments follow below: 

Summary of the paper

The present study investigates the association between sleep and health functioning and premature exit from the work force. The paper has a strong methodological focus. The topic is not novel, but the authors uses a different methodological approach than other studies having investigated this association. Most studies have used conventional risk factor modelling with Cox regression, but the present study uses repeated measures latent class analyses (RMLCA), which can identify specific groups within the sample. The study population is employees of the City of Helsinki in their midlife period. Three health surveys were conducted at different time points and the study utilizes this data. By employing the RMLCA methodology, separate subgroups according to degree of sleep problems and health functioning are identified. In addition to examining the association using the RMLCA approach, the study also aims to assess whether this approach is more predictive than a traditional Cox model. The findings showed an increased risk of premature exit of the workforce for the groups with problems of sleep and functioning compared to the group with good sleep and functioning. Furthermore, compared with to traditional risk factor based approaches, RMLCA could be a more powerful method for quantifying the role of sleep and health functioning as risk factors for premature exit of the work force.

General impression

The paper is generally a very interesting and well conducted piece of research covering an important topic. Although the topic is not novel, I do find that it adds value to the current knowledge base by employing a different methodological approach. However, I find that several improvements should be made before the paper is ready for publication.  

Regarding presentation of the aims of the paper: Although it is possible to identify the aims of the paper by reading it in its present state, I still find that the aims come across as somewhat unfocused and could presented in a way that is more clear to the reader.

Furthermore, the paper could benefit from some restructuring as I find that some of the text should be moved to other chapters. E.g. the second paragraph in 4. Discussion is very focused on results and could preferably be moved the results chapter.

Language

  • I find that this paper would benefit from language washing, or that the authors improve the language themselves. Correction of typos, grammatical errors, and better sentence structure would greatly improve the overall impression of the paper.
  • Several times in the paper words are written in quotation marks; I suggest that you rewrite the relevant sentences to avoid the extensive use of quotation marks.

Formatting

  • Using more sub-headings would make the paper easier to read. E.g. under chapter 2. Measures, the sub-headings “I. Sleep variables” and “II. Health Related Functioning” should preferably be e.g. underlined in order to make them stand out from other text. Similar format changes could also be done other places were relevant.

Abbreviations

  • Please always write all abbreviations fully the first time; e.g. OSA is not defined in the Introduction. Also, after you have defined an abbreviation, you should continue to use it; on page 4 you define vitality as VI in the third line, but some lines below you still write “vitality” in full instead of using the defined abbreviation VI.

References

  • Please be consistent when using references in the text. Mostly, references are presented as numbers in brackets, but sometimes as numbers without brackets.

Abstract

Instead of starting the abstract with the aim, you may start the abstract with a sentence or two about the background of the article.

It should be clear to the reader already from the abstract that this study both assesses the association of sleep/functioning and workforce exit and compares it to more a traditional methods.

I do not find the sentence “In two models we identified a fifth subgroup…” well integrated. Please explain further.

As a conclusion you write that “longitudinal  person-oriented approaches may be a more powerful method…”. It is previously stated in the abstract that the aim is “ to examine the pattern of interactions between sleep and health functioning characterizing homogeneous subgroups of employees and to determine whether the risk of the premature exit from paid employment differs between these subgroups over ten years.” I can’t see that the authors have concluded on the stated aims of the study; please modify the conclusion to conclude with respect to the aims.

The statement “These groups might be targeted for interventional studies” seems a little out of place here. It is a very interesting point, but I suggest that you rather move it to the discussion and expand upon it there. Perhaps in a paragraph regarding future research.

Introduction

In the Introduction, you may work to present the aim of the study more clearly and make sure that it is consistent with how the aims is presented in other parts of the paper.

Could you also present the rationale for identifying the sub-groups combining sleep and functioning more clearly. Would it not be interesting to look at sleep and health as separate problems as well?

Materials and methods

A list of covariates to be included in the analyses is presented. Although you write that these variables are pertinent, I recommend that you also include a theoretical rationale for why exactly these variables should be included.

Perhaps include an example of what is meant by “semi-professionals” for the variable occupational class.   

The presentation of the steps of the statistical analysis could preferably presented more concisely in order to make it easy for the reader to follow. I find this important since the methods that are used are not common for studying the topic of the present paper.

Results

Please also include a paragraph describing the data in the text rather than just referring to Table 1.

Table 1: By indicating each different variable with e.g. indentation, bold, or italics etc., the table will be easier to read.  

Second paragraph, page 14 (“It is obvious that “unwanted”…”): This paragraph will read better if you rewrite it so you do not have to repeat the phrase “the next strongest” at several occasions.  

Discussion

The first two paragraphs in the discussion may be rewritten to present a shorter and more to-the-point summary of the results and some aspects of the methodology. The way it is written now, the first paragraph seems to fit better in the methods chapter and the second paragraph seems to fit better in the results chapter.

In the end of the first paragraph, it is unclear to me why the “person-oriented approach indicated that the decisive factors for the effect of mental health problems is their chronicity”. The text leading up to this conclusion could be rewritten to get the message across more clearly.

I suggest that you include a separate sub-chapter on strengths and limitations in the discussion rather than including it under “Methodological considerations”.

Some of the information in “Methodological considerations” is better suited for the results chapter (e.g. prevalence, indicators of reliability etc.). I suggest that you move some of this to Results and rather discuss the main concepts without referring to numbers at such a large extent in “Methodological considerations”.

In your sample, the proportion of women is a lot larger than the proportion of men (82 % vs. 18 % at baseline). I cannot find that you have discussed this point. Do you think this could be of relevance for your findings?

I also miss a more thorough discussion on why the person-centered approach is better than more traditional approaches. Could it be that neither is better than the other in all circumstances, but rather that the best method depends on the goal of the study? Perhaps also the two methods may supplement each other?

Conclusions

I find the conclusion a bit unfocused since it contains both methodological aspects like the first sentence on “person-oriented repeated measures…” and aspects about future research (the last 5-6 lines starting with “It would be very…”). I suggest to only focus on what can be concluded from the present study in Conclusions. Perhaps you also may include a bit on implications of the study. The ideas for future research could preferably be moved to the discussion.

Author Response

Reviewer 1

Thank you for the opportunity to review this interesting paper. My comments follow below: 

Summary of the paper

The present study investigates the association between sleep and health functioning and premature exit from the work force. The paper has a strong methodological focus. The topic is not novel, but the authors uses a different methodological approach than other studies having investigated this association. Most studies have used conventional risk factor modelling with Cox regression, but the present study uses repeated measures latent class analyses (RMLCA), which can identify specific groups within the sample. The study population is employees of the City of Helsinki in their midlife period. Three health surveys were conducted at different time points and the study utilizes this data. By employing the RMLCA methodology, separate subgroups according to degree of sleep problems and health functioning are identified. In addition to examining the association using the RMLCA approach, the study also aims to assess whether this approach is more predictive than a traditional Cox model. The findings showed an increased risk of premature exit of the workforce for the groups with problems of sleep and functioning compared to the group with good sleep and functioning. Furthermore, compared with to traditional risk factor based approaches, RMLCA could be a more powerful method for quantifying the role of sleep and health functioning as risk factors for premature exit of the work force.

General impression

The paper is generally a very interesting and well conducted piece of research covering an important topic. Although the topic is not novel, I do find that it adds value to the current knowledge base by employing a different methodological approach. However, I find that several improvements should be made before the paper is ready for publication.

Response:  Thank you for this very positive feedback on our study, and the appreciation to the method used.

Regarding presentation of the aims of the paper: Although it is possible to identify the aims of the paper by reading it in its present state, I still find that the aims come across as somewhat unfocused and could presented in a way that is more clear to the reader.

Response: We have clarified the aims to be more focused, and revised their presentation as suggested. We started the aims using a new paragraph, and also clearly distinguished between the 2 aims, one being about the associations and applying a new method in this area, and the second one providing a tutorial, i.e., longer than typical explanations of the RMLCA models to help others apply them in their research questions. In brief, this study was partly a methods paper and partly a tutorial. We hope the current way of presenting the aims is clearer (please see last paragraph of the introduction).

Furthermore, the paper could benefit from some restructuring as I find that some of the text should be moved to other chapters. E.g. the second paragraph in 4. Discussion is very focused on results and could preferably be moved the results chapter.

Response: We have tightened text in the discussion, not to repeat the results. The results are presented already in the results section but in a different way. Thus replacing them into the result section, as suggested, is not feasible. As we have many models, we initially had the intention to summarize some key findings in the discussion. However, all the figures (HRs etc.) are now omitted, and the paragraph is notably shorter.

Language

  • I find that this paper would benefit from language washing, or that the authors improve the language themselves. Correction of typos, grammatical errors, and better sentence structure would greatly improve the overall impression of the paper.
  • Several times in the paper words are written in quotation marks; I suggest that you rewrite the relevant sentences to avoid the extensive use of quotation marks.

Response: We have made our best efforts to revise the language, and correct any errors. We have also rewritten all the sentences to be without a quotation mark, as suggested. We hope the revised version reads better. Those marks were used for names of the latent classes or direct citations of the names of the measures or survey items.

One of the authors is a native English speaker, and he has particularly helped make the text more understandable and helped improve the readability of the text.

Formatting

  • Using more sub-headings would make the paper easier to read. E.g. under chapter 2. Measures, the sub-headings “I. Sleep variables” and “II. Health Related Functioning” should preferably be e.g. underlined in order to make them stand out from other text. Similar format changes could also be done other places were relevant.

Response: We have added more sub-headings, as suggested, and revised the sub-headings to stand out better from other text e.g. as underlined. We have also added extra space and lines to make the paper easier to read. New headings are shown in red font.

Abbreviations

  • Please always write all abbreviations fully the first time; e.g. OSA is not defined in the Introduction. Also, after you have defined an abbreviation, you should continue to use it; on page 4 you define vitality as VI in the third line, but some lines below you still write “vitality” in full instead of using the defined abbreviation VI.

Response: We have opened up all the abbreviations fully, when they are first used, and then used the abbreviations consistently.

References

  • Please be consistent when using references in the text. Mostly, references are presented as numbers in brackets, but sometimes as numbers without brackets.

Response: We have checked and revised the format of the references, as suggested.

Abstract

Instead of starting the abstract with the aim, you may start the abstract with a sentence or two about the background of the article.

It should be clear to the reader already from the abstract that this study both assesses the association of sleep/functioning and workforce exit and compares it to more a traditional methods.

I do not find the sentence “In two models we identified a fifth subgroup…” well integrated. Please explain further.

As a conclusion you write that “longitudinal  person-oriented approaches may be a more powerful method…”. It is previously stated in the abstract that the aim is “ to examine the pattern of interactions between sleep and health functioning characterizing homogeneous subgroups of employees and to determine whether the risk of the premature exit from paid employment differs between these subgroups over ten years.” I can’t see that the authors have concluded on the stated aims of the study; please modify the conclusion to conclude with respect to the aims.

The statement “These groups might be targeted for interventional studies” seems a little out of place here. It is a very interesting point, but I suggest that you rather move it to the discussion and expand upon it there. Perhaps in a paragraph regarding future research.

Response: We have revised the abstract following the suggested points. This is a paper with rather detailed methods and results and it is difficult to fit in all the details within the limits of the abstract word count. We have tried to highlight the main points in the abstract, and then other details are in the full paper. As this is an open access journal, all have access to any detail they are interested in, unlike in some other papers where abstract is the only accessible source. However, our main aim is not to compare this method to traditional methods, although we report such comparisons. We preferred not to highlight that in the abstract. Instead, an additional aim was to make a tutorial, providing enough details to support other researchers to apply the method in addressing their research questions.

We have, however, deleted the sentence “In two models we identified a fifth subgroup of improving functioning with improving or normal sleep” as it was mistakenly forgotten there. Initially we identified also some 5 class solutions in two RMLCA models. However, because of mathematical reasons and weak evidence suggesting their acceptance, we rejected them and in order to avoid speculation, and did not include them in the final MS, or any text about them.

We have also deleted the sentence about interventional studies. However, the conclusions are indeed in line with aims. Identifying latent groups (development of sleep and functioning) means that we could find risk groups for early exit, and intervention studies could then use this information. We have added a new sentence and reformulated the initial sentence and hope that the abstract and the revised manuscript better explains the point of these models and justifies our conclusions.

Introduction

In the Introduction, you may work to present the aim of the study more clearly and make sure that it is consistent with how the aims is presented in other parts of the paper.

Could you also present the rationale for identifying the sub-groups combining sleep and functioning more clearly. Would it not be interesting to look at sleep and health as separate problems as well?

Response: We have revised the aim, as suggested. We have also clarified and amended our rationale for combining sleep and functioning in the introduction (please see red text in the last paragraphs of the introduction). It is true that they are interesting also as separate predictors, but unfortunately it would be too much for a paper to separately model RMLCA for sleep, and for functioning, and for their combinations. Also, there is previous work on both of these predictors separately, but we aimed to focus on them jointly. Studies looking at either sleep or functioning would miss the contributions that they have together, and that e.g. poor sleep alone is not as harmful as when it is combined with poor functioning.

Materials and methods

A list of covariates to be included in the analyses is presented. Although you write that these variables are pertinent, I recommend that you also include a theoretical rationale for why exactly these variables should be included.

Response: We have aimed to clarify rationale for covariates. As they are many, a detailed, theory-based explanation would take a lot of space, and also shift the focus of the introduction to a ‘side track’. The inclusion of covariates is based on a large number of previous studies on sleep, functioning and premature exit from the workforce. We have cited several papers in the introduction that have used similar background factors [please see ref 4, 5, 6, 11, 14]. This can be seen as a justification for their inclusions. As this paper is not able to focus on e.g. mechanisms, and also does not aim to e.g. explain the associations between predictors and outcome, the role of these covariates is minor. They are needed as they may affect the associations, but we do not have an aim to use them as explanatory factors, or focus on their own effects per se, when more justification and theoretical rationale could be warranted.

Perhaps include an example of what is meant by “semi-professionals” for the variable occupational class.   

Response:

We have opened up the categories of occupational social class in a bit more detail. This is a covariate and it has been in many hundreds of studies. Studies which have used it as a key determinant provide more details. The City of Helsinki has hundreds of different occupational titles, and these have merged into 4 occupational social classes, alongside the guidelines by the Statistics Finland.

Managers have subordinates and do managerial/administrative work. They typically have a university degree. Professionals, including other upper white-collar employees, such as teachers and doctors, also have a university degree, but do professional work and they typically do not have subordinates. Semi-professional jobs require a college-level qualification or they can include supervisory but also routine tasks, which are characterized by having less autonomy. Semi-professionals in the municipal sector include e.g. nurses, foremen and technicians, and other intermediate level white-collar employees. Routine non-manuals and manual positions require vocational training or they have no specific qualifications, with routine-non manuals being employed in non-supervisory clerical or non-manual tasks, and manual workers in e.g. transportation or cleaning.

The presentation of the steps of the statistical analysis could preferably presented more concisely in order to make it easy for the reader to follow. I find this important since the methods that are used are not common for studying the topic of the present paper.

Response: We have tried to clarify the methods. As our aim was to also make this partly a tutorial, we feel it is necessary to give enough details on the method. In particular, if the method is new to the readers, they need more information to be able to repeat the method to address their research questions. We also received requirements to provide more details, making the revisions contradictory.

Results

Please also include a paragraph describing the data in the text rather than just referring to Table 1.

Response: We have amended the description of the table 1. Please find new added text under the section 3.1. We feel it is best not to repeat too many details that are in the table, as this is the guideline. Also we feel it is better to save enough space for the models and the analytical steps, rather than the very descriptive parts.

Table 1: By indicating each different variable with e.g. indentation, bold, or italics etc., the table will be easier to read.  

Response: We agree that highlighting some variables would be useful, but this journal has advised against using such highlight (removed italics and other formats from the previous manuscript, too).

Second paragraph, page 14 (“It is obvious that “unwanted”…”): This paragraph will read better if you rewrite it so you do not have to repeat the phrase “the next strongest” at several occasions.  

Response: We have rewritten the paragraph, as suggested, omitting the words “the next strongest” and merging the sentences to make the paragraph read better.

Discussion

The first two paragraphs in the discussion may be rewritten to present a shorter and more to-the-point summary of the results and some aspects of the methodology. The way it is written now, the first paragraph seems to fit better in the methods chapter and the second paragraph seems to fit better in the results chapter.

Response: We have rewritten the paragraphs, as suggested, for them to better fit in the discussion section. The 2nd paragraph is now notably shorter and we have omitted all figures to make the paragraph more suitable for discussion. For the first paragraph, we added a subheading to help readers understand that it is about the main findings of the study.

In the end of the first paragraph, it is unclear to me why the “person-oriented approach indicated that the decisive factors for the effect of mental health problems is their chronicity”. The text leading up to this conclusion could be rewritten to get the message across more clearly.

Response: We have revised the paragraph to be clearer, and to better lead to our given conclusion. We could not find it in the end of the first paragraph but it is in the section 4.2 Interpretation. The chronicity means that problems were repeatedly found across all three waves, i.e. over the 10 to 12 year follow-up, and having the repeated problems had the strongest associations with early exit from workforce. We have clarified the sentence and added that the stability of mental health problems was indicated using repeatedly collected data (repeated measurements).

I suggest that you include a separate sub-chapter on strengths and limitations in the discussion rather than including it under “Methodological considerations”.

Response: We have divided the methodological considerations into two separate section, as suggested (strengths and limitations). These sometimes partly coincide, and are not fully separate. We have added a new paragraph under the title ‘strengths’.

Some of the information in “Methodological considerations” is better suited for the results chapter (e.g. prevalence, indicators of reliability etc.). I suggest that you move some of this to Results and rather discuss the main concepts without referring to numbers at such a large extent in “Methodological considerations”.

Response: We have revised the methodological considerations, by adding sub-headings as well as a new paragraph of strengths. One might consider that information could fit into different parts of the manuscript, but here they are discussed from different angels. We felt the logic and structure of the paper was better when we retained the information in the methodological considerations.

In your sample, the proportion of women is a lot larger than the proportion of men (82 % vs. 18 % at baseline). I cannot find that you have discussed this point. Do you think this could be of relevance for your findings?

Response: We have amended the manuscript to better state that the proportion of women corresponds to that of the target population (please see revised limitations section under 4.3). Thus, the Finnish municipal sector (in general) is (very) female dominated (70-80% depending on age group). It may have a relevance e.g. re generalizability of the findings to other sectors. However, although prevalence of poor sleep and functioning differ for women and men, their patterning can be similar and also there is no reason to assume why poor sleep and poor functioning would not similarly be associated with early exit among women and men. The same has been shown using variable-oriented methods.

I also miss a more thorough discussion on why the person-centered approach is better than more traditional approaches. Could it be that neither is better than the other in all circumstances, but rather that the best method depends on the goal of the study? Perhaps also the two methods may supplement each other?

Response: We agree with your points, and have clarified the manuscript accordingly. We are not saying that either method is superior but rather may address different questions and exactly complement each other (please see revised Conclusions section). Other points about comparisons between person-oriented and variable-oriented methods are covered e.g. in section 3.5 and through the discussion.

Conclusions

I find the conclusion a bit unfocused since it contains both methodological aspects like the first sentence on “person-oriented repeated measures…” and aspects about future research (the last 5-6 lines starting with “It would be very…”). I suggest to only focus on what can be concluded from the present study in Conclusions. Perhaps you also may include a bit on implications of the study. The ideas for future research could preferably be moved to the discussion.

Response: We have clarified the conclusions, as suggested. We feel it is necessary to highlight person-oriented methods, as that was our aim. However, we have moved ideas for future research into the discussion (end of the section 4.2).

Submission Date

30 December 2020

Date of this review

19 Jan 2021 21:48:58

Reviewer 2 Report

The article reviews the "Associations of Sleep and Health Functioning with Premature Exit from Work: A Cohort Study with a Methodological Emphasis". As the authors mention - it is concise and clinically oriented, however, it lacks the following points:

- editors notice a mistake in reference: "Sickness allowance due to somatic conditions is a weaker risk factor.8"

It's not clear what kind of inclusion and exclusion criteria were used? have You excluded patients with primary sleep disorders like insomnia, obstructive sleep apnea syndrome; circadian rhythm sleep-wake disorders; shift workers? How about mood disorders especially depression? 

 Have you considered to perform separate analysis to compare subgroups of workers due to seasons in which the survey was performed? A short day with sun activity Is considered as one of the risk factors of depression, that affect daily activity. It might be interesting to compare those subgroups due to four seasons. 

Author Response

Reviewer 2

Comments and Suggestions for Authors

The article reviews the "Associations of Sleep and Health Functioning with Premature Exit from Work: A Cohort Study with a Methodological Emphasis". As the authors mention - it is concise and clinically oriented, however, it lacks the following points:

- editors notice a mistake in reference: "Sickness allowance due to somatic conditions is a weaker risk factor.8"

Response: Thank you, we have corrected the reference.

It's not clear what kind of inclusion and exclusion criteria were used? have You excluded patients with primary sleep disorders like insomnia, obstructive sleep apnea syndrome; circadian rhythm sleep-wake disorders; shift workers? How about mood disorders especially depression? 

Response: We have clarified the inclusion and exclusion criteria in the revised methods section. We focused on a sample of midlife employees (i.e. not a sample of patients) of the City of Helsinki, and their insomnia symptoms, functioning and early exit from the work force. We did not have any information on their obstructive sleep apnea syndrome or circadian rhythm sleep-wake disorders, and also survey based insomnia symptoms are not specific to insomnia. We have information about shift work, and about a fifth are doing shift work. We did not exclude them. Also we did not have information on clinical depression. It is also a bit problematic when focusing on insomnia symptoms and their development over time, and insomnia may also precede depression.

Have you considered to perform separate analysis to compare subgroups of workers due to seasons in which the survey was performed? A short day with sun activity Is considered as one of the risk factors of depression, that affect daily activity. It might be interesting to compare those subgroups due to four seasons. 

Response: Thank you for this suggestion. It would be interesting to focus on the seasons but unfortunately that cannot be done in this study. We followed-up people between 10 to 12 years, and most surveys were collected during autumns. However, a small numbers could return their surveys late, e.g. during early winter, and the first wave was collected in spring. In all, we do not have data to focus on seasons, but most participated in autumn. We have mentioned this in the revised discussion.

Submission Date

30 December 2020

Date of this review

18 Jan 2021 14:54:03

Reviewer 3 Report

This study deals with the question of whether health functioning and insomnia symptoms predict early exit in the general workforce. The number of subjects was very large, and the results reflect long term follow-up.

This manuscript presents an association of some insomnia symptoms and quality of life measures with early exit, which can be important for public health. However, I still have serious concerns on several issues in this paper.

Major comments:

  1. In the Methods section, it is stated that "However, it has been argued that MCS and PCS … to avoid misleading or inaccurate conclusions".
    Most of the epidemiological studies using SF-36 to date have simply applied the PCS and MCS scores. I thought it would be preferable to do the same analysis using the MCS and PCS total scores, at least as a sensitivity analysis. Have these been performed?

  1. It seems that JSQ and SF-36 were examined in each of Phases 1, 2, and 3 for the same subjects. It was not clear to me how the results from these three studies were assigned to LC 1 to 4. This may be due to my lack of understanding. Is it the RMLCA that integrates each of these three scores? If possible, please add some more information to make it clearer for the reader.

  1. Related to the above, were measures of insomnia and QOL at 2012 used to assign the LC in this study? For some subjects, measures of insomnia symptoms and QOL after retirement may have been applied. A number of studies have already shown that insomnia symptoms worsen after retirement (see https://doi.org/10.1007/s40675-018-0132-5), which may have influenced the LC classification in this study. In that case, one of the main outcomes of the present study, that insomnia symptoms predict early exit, will be questioned.

  1. In the Results section, it is stated that "However, because of obvious over adjustment, self-reported sleep duration was not included among covariates." However, this may reflect the fact that short or long sleep is strongly associated with early job turnover. There are several cross-sectional and longitudinal studies showing that short or long sleep is associated with QOL in the general population, as shown below.

https://doi.org/10.1016/j.sleep.2020.10.012

https://doi.org/10.1016/j.sleep.2016.03.008

https://doi.org/10.1371/journal.pone.0187275

In addition, the association between sleep duration and insomnia symptoms to early exit has been already reported by the same group in a study using presumably the same cohort (https://doi.org/10.5271/sjweh.3269, cited as [14] in the manuscript). However, there is not much mention of this study in this manuscript, which raises the suspicion that the authors may have arbitrarily excluded sleep duration from the covariates.

Minor comments:

  1. It is mentioned as "The model solution was rejected because of interpretational difficulties" in footnote 1 of Table 3.

I could not understand the reason. Please explain a more detailed reason.

  1. In Table 3, there is a notation "Bodily pain (PB)". Please fix it.

  1. There were some references that were not enclosed in “[]”. Please fix them.

Author Response

Reviewer 3

Comments and Suggestions for Authors

This study deals with the question of whether health functioning and insomnia symptoms predict early exit in the general workforce. The number of subjects was very large, and the results reflect long term follow-up.

This manuscript presents an association of some insomnia symptoms and quality of life measures with early exit, which can be important for public health. However, I still have serious concerns on several issues in this paper.

Response: Thank you for your comments.

Major comments:

  1. In the Methods section, it is stated that "However, it has been argued that MCS and PCS … to avoid misleading or inaccurate conclusions".
    Most of the epidemiological studies using SF-36 to date have simply applied the PCS and MCS scores. I thought it would be preferable to do the same analysis using the MCS and PCS total scores, at least as a sensitivity analysis. Have these been performed?

Response: It is true that despite criticism, many studies use PCS and MCS. As a sensitivity analysis, also PCS and MCS models were analyzed and the results were in line with SF-36 subscale analyses. We mention this in the revised ms, in a paragraph in the end of the section 3.2. The results and a broad picture were similar (patterns of change using the PCS and MCS), but we preferred to use the subscales that are more specific, to get a clearer picture on the development of sleep and functioning. PCS and MCS include all the items and then we do not have as clear or specific information on the risk groups, as we have when using the subscales. As there are already 12 different models (different combinations of physical and mental functioning and sleep problems), it is not meaningful to add 6 more figures in the paper. Other, more methodological reasons are given in the manuscript, with references.

  1. It seems that JSQ and SF-36 were examined in each of Phases 1, 2, and 3 for the same subjects. It was not clear to me how the results from these three studies were assigned to LC 1 to 4. This may be due to my lack of understanding. Is it the RMLCA that integrates each of these three scores? If possible, please add some more information to make it clearer for the reader.

Response: RMLCA can be seen as integrating measures in all time points. Classes of the RMLCA could be seen like developmental dual trajectories, showing the development of sleep and functioning over time. They describe the changes that happen during the follow-up (10-12 years) in sleep and functioning. Each latent RMLCA class comprises individuals who are likely to have similar development (similar response pattern over the follow-up) in their sleep and functioning over the follow-up (e.g. all members of LC 4 report consistently poor sleep and poor functioning at each phase during the follow-up and members of LC 1 report consistently absence (very low) of problems). We have already quite detailed descriptions of the models, but we hope the revised version is clearer also in this respect.

Finally, we added an example to illustrate the mathematical challenges of this modelling more concretely. We currently have three measurement points (phases 1, 2 and 3) and two dichotomized indicators (sleep and functioning). Thus, there are altogether 64 possible response-patterns or ’trajectories’ (23x23=64). All these response patterns form a matrix of response patterns, or an array of response patterns. This equals to the number of cells in the contingency table that we analyze. Thus, if we add for example even one additional indicator in our model, the number of possible response patterns increase to 512 (23x23x23=512), and adding a fourth one, the number is as high as 4096 (23x23x23x 23=4096), etc. Including 7 indicators into the model at the same time would mean that we would end up with more than 2 million possible response patterns to analyze (7 dichotomous variables in 3 time points, i.e. 23x23x23x23x23x23x23= 2 097 152). If the indicator variables had more than two categories, the numbers would obviously be even a lot higher, as the current calculations above are all based on dichotomous variables.

  1. Related to the above, were measures of insomnia and QOL at 2012 used to assign the LC in this study? For some subjects, measures of insomnia symptoms and QOL after retirement may have been applied. A number of studies have already shown that insomnia symptoms worsen after retirement (see https://doi.org/10.1007/s40675-018-0132-5), which may have influenced the LC classification in this study. In that case, one of the main outcomes of the present study, that insomnia symptoms predict early exit, will be questioned.

Response: We would like to emphasize that sleep and functioning were not used as predictors, and no causal assumption are warranted. Instead, we examine and refer to only associations between the latent groups and early exit. Thus, we know how sleep and functioning develop over time, and have assigned each participant in the latent group where they most likely belong to. Each class is as homogenous as possible in terms of development and sleep during the follow-up. Then we examine if the latent group membership is associated with an early exit during the follow-up. This aim can be addressed with the current design, while another type of study could examine causality. The results show, how latent group membership associated with the hazard of early exit, which occurs during the follow-up. We have clarified this further in the revised discussion, under the section 4.3, first paragraph.

The review study cited (Myllyntausta & Stenholm, 2018) on the contrary shows that after statutory retirement (age based normal retirement), insomnia symptoms typically decrease (not increase). For those people who retired based on the age, it is possible that their sleep may have improved. For early exit, it is less clear what may happen with their sleep. However, when the focus is on identifying latent groups in the development of sleep and functioning, this is less crucial. Moreover, we show that the development is rather stable, or that those with poor sleep and functioning at baseline, when still working, are likely to have poor sleep and functioning throughout the follow-up, and likewise good sleep and good functioning remain stable, etc. We focused on employed people (at baseline) and their early exit, and whether patterns of change in their sleep and functioning over follow-up (the identified RMLCA classes) are associated with early exit. Changes among those who continue working (do not retire during follow-up) and then those who retire due to old age can be seen as controlling variables, when we assess the hazard of early exit.

  1. In the Results section, it is stated that "However, because of obvious over adjustment, self-reported sleep duration was not included among covariates." However, this may reflect the fact that short or long sleep is strongly associated with early job turnover. There are several cross-sectional and longitudinal studies showing that short or long sleep is associated with QOL in the general population, as shown below.

Response: It is true that sleep duration is associated with functioning, but it also strongly associates with insomnia symptoms. Thus, adjusting for sleep duration would create a notable risk of over-adjustment (adjusting the effect of insomnia with “insomnia”). As sleep duration is also associated with the outcome, one has to choose between the two indicators, insomnia or sleep duration. They bring into the model largely the same information and adding one into the model, when the other already is there, does not improve the model. Sleep duration is, however, more heterogeneous (e.g. reasons to short and long sleep vary from physiological to voluntary sleep restriction, and different sleep disorders) than insomnia, and we preferred to have a focus on insomnia-related symptoms. There is also a technical reason, as we need more categories for sleep duration, to distinguish between short, ‘average’ and long sleep. Adding a category makes the RMLCA model a lot more complex, as we also have 3 time points (Instead of 64 [23x23] possible response patterns we would have 216 [23x33] response patterns to analyze). Therefore, after careful consideration and modelling, we omitted those indicator variables that did not add information in the model. Instead of using it as an indicator, we analyzed the association of sleep duration with latent groups based on insomnia symptoms and health functioning. Consequently, the role of sleep duration in premature exit from work was not neglected in our analyses.

We have clarified its role in the revised manuscript, in the section 2.2, under Sleep variables.

In addition, the association between sleep duration and insomnia symptoms to early exit has been already reported by the same group in a study using presumably the same cohort (https://doi.org/10.5271/sjweh.3269, cited as [14] in the manuscript). However, there is not much mention of this study in this manuscript, which raises the suspicion that the authors may have arbitrarily excluded sleep duration from the covariates.

Response: The study cited above is very different from the current study. Firstly, it, like many other studies of the field, is based on variable-oriented analyses. We have clarified the description in the revised introduction. The study also did not focus on functioning, and did not at all focus on how sleep and functioning develop over time, i.e., what are the patterns of change that we identified here. Please also see our response above and in the revised manuscript regarding reasons to omit sleep duration in this particular design and in the person-oriented method, which requirements differ from the prior studies.

Minor comments:

  1. It is mentioned as "The model solution was rejected because of interpretational difficulties" in footnote 1 of Table 3.

I could not understand the reason. Please explain a more detailed reason.

Response: We have clarified this (please see Table 3, footnote 1). In the RMLCA models, if there is no meaningful interpretation for the model (for the latent classes), or how the classes are distinct with respect to the phenomenon examined, such a classification cannot be considered. There is statistical significance to consider, but also there has to be an interpretation for the classes identified. In other words, if a model does not add any new information or deepen our understanding about the patterns of changes in sleep and functioning, such a model is rejected. Statistically, some models may fulfil mathematical criteria but still be meaningless, for example a class could be really small, or two classes could follow really similar development.

  1. In Table 3, there is a notation "Bodily pain (PB)". Please fix it.

 Response: Thank you, this has been corrected.

  1. There were some references that were not enclosed in “[]”. Please fix them.

Response: We have fixed the errors in the references.

Submission Date

30 December 2020

Date of this review

12 Jan 2021 08:10:39

Round 2

Reviewer 3 Report

The authors have adequately responded to the comments of the reviewer in this revision.
I apologize for my mistakes and lack of understanding. Thank you for the opportunity to review this interesting paper.